
# Assessing flood hazard changes using climate model forcing

David P. Callaghan[1] and Michael G. Hughes[2,3]

[1]School of Civil Engineering, The University of Queensland, Brisbane, QLD, Australia,
[2]Science Economics and Insights Division, NSW Department of Planning and Environment, Australia,
[3]School of Earth Atmospheric and Life Sciences, University of Wollongong, North Wollongong, NSW, Australia.

*Correspondence to*: David P. Callaghan (dave.callaghan@uq.edu.au, davepcallaghan@gmail.com)

**Abstract.**

A modelling framework for using regional climate projections to assess flooding hazard has been developed and applied to the Gwydir River (catchment 26,600 km$^2$ and floodplain 8,100 km$^2$), NSW, Australia. The model framework uses NSW and

ACT Regional Climate Modelling version 1.5 projections combined with computationally efficient hydrologic and hydraulic models. While requiring model management and high-performance computing resources, the modelling framework successfully processed 18 regional climate projections into flood projections. Specifically, a six-member set of climate model combinations simulating a historical period (1950-2006) and a future period (2006-2100) under two global emission pathways (RCP4.5 and RP8.5) were used to predict flood depth and speed. In total, 1,470 continuous years were simulated at

hourly time step. These flood (depth and speed) projections were analysed to assess the flood hazard changes under future climate scenarios by estimating changes in the annual probability of occurrence of a range of flood hazard classes. The six-member ensemble indicates flood hazard in the Gwydir Valley will decrease in the short, medium and long term. There are also cases within the ensemble which includes increases in all non-safe flood hazard classification while decreasing the safe flood hazard classification.

Short summary as a 500-character (incl. spaces) non-technical text.

Regional climate change modelling output of rainfall and soil moisture have been used to estimate flooding hazard across the entire Gwydir River floodplain.

# 1 Introduction

Climate change potentially includes changes in temperature, evaporation, rainfall, and their seasonal patterns. Changes in rainfall patterns translate to changes in flooding extent, duration, and strength (i.e. flood hazard). Preparing for potential future changes in flood hazard can require significant lead times, thus it is critical to incorporate climate change information into flood hazard risk assessment and adaptation planning. One way to investigate the nature of potential future flooding involves climate model outputs being converted to hydrodynamic outputs (flow depth and speed as a function of time), but

this is not a trivial task and there is no general agreement on an approach. For example, future climate-related changes in the



fitted distributions to channel flow estimates have been evaluated using stochastic methods, water balance modelling and change factors (Delgado et al., 2014; Hirabayashi et al., 2013; Smith et al., 2014a). On the other hand, the direct application of climate model outputs has been discouraged by some (Cloke et al., 2013; Prudhomme et al., 2010). Nevertheless, some accelerated models for converting flow rate into floodplain inundation show promise for converting regional-scale climate
model outputs into continuous flood dynamics for hazard assessment on large and complex floodplains (e.g., Bates et al., 2010; Falter et al., 2013; Ghimire et al., 2013; Lhomme et al., 2009).

Assessing the future flood hazard under climate change directly (i.e. from hazard = depth × speed) at regional or jurisdictional scales requires the ability to simulate river flows and floodplain inundation at hourly (or better) time scales
over many decades and across large areas. The necessary computational efficiency can potentially be achieved by a variety of physics-based approaches including dynamic wave, partial inertial wave, diffusive wave and kinematic wave models (e.g., Montanari et al., 2009; Bates et al., 2010; De Roo et al., 2000; Miller, 1984). These involve simplifying the physics that are simulated together with a reduction in detail for one or two of the flow dimensions. For example, the computationally efficient LISFLOOD-FP offers options to implement as dynamic wave, partial inertial wave or kinematic wave depending on
what the environment being modelled demands (Lhomme et al., 2010; Bates et al., 2010; Bates et al., 2005). Decisions are therefore required on which physical processes can safely be ignored in the river environment of interest. Alternatively, there are computationally efficient rules-based models which involve a set of rules that mimic continuity and kinematic limits (e.g., Guidolin et al., 2016). The best choice of these two model approaches for undertaking a flood hazard assessment under future climate change optimises the trade-off between model accuracy and computational effort with obtaining the necessary
flood outputs to calculate hazard.

Performing hazard risk assessments and developing adaptation strategies for hazards under future climate change generally requires regional-scale (or better) climate projections. This involves refinement of global climate models through either statistical down-scaling (e.g., Wilby et al., 1998; Schmidli et al., 2006; Timbal and Jones, 2008) or dynamical down-scaling
(e.g., Laprise, 2008; Giorgi, 2006; Ekström et al., 2015). The Australian NSW and ACT Regional Climate Model (NARCliM) is one example of this approach and used dynamical downscaling of a global 50 km model grid to a regional 10 km model grid (Evans et al., 2014; Nishant et al., 2021). Climate models represent the distribution of weather and as such, comparisons between climate model predictions and historical measurements are possible by comparing their distributions but not by comparing specific historical events. Comparing distributions requires a balance between a measurement and
model record long enough for such distributions to be appropriately defined while being short enough to limit non-stationary impacts from the changing climate. For parameters such as daily temperature or average rainfall, a 20-year period is suitable given there are many rainfall events per year and every day has a maximum temperature (near continuous variable). For parameters with rarer occurrences, such as floodplain inundation, defining their distribution becomes increasingly more marginal. For example, defining changes in flood inundation that is exceeded every 100 years using a 20-year simulation





period comes with considerable uncertainty. However, we may be able to usefully compare relevant measurements and model predictions for more frequent events, such as the annual flood hazard classes.

The purpose of this paper is to describe the successful application of a modelling framework developed to convert climate model projections to hydrodynamic outputs, which were then used to assess future changes to present-day regional flood

hazard. We demonstrate the utility of the approach by applying it to the Gwydir River, a large valley-floodplain system located in the northern Murray-Darling Basin, Australia. After reviewing candidate numerical models, new methods for driving a ROR-style hydrology model and the LISFLOOD-FP hydraulic model with climate projections are described, driven by NARCliM1.5 climate projections as an example. Projected future regional flood inundation extents and the spatial distribution of flood hazard are presented for two global emission pathways (RCP4.5 and RCP8.5). Challenges associated

with spatial and temporal sparsity in floodplain inundation and applying conventional extreme value distributions to evaluate future flood exceedance probabilities are discussed. These confound efforts to answer the question – will present-day flood hazard change under future climate projections – and we provide a new approach to answering that question.

## 2 Methods

The objectives in converting climate model outputs to inundation estimates were: i) develop a method for manipulating

NARCliM 1.5 hydrological variables for application in rainfall-runoff routing models, ii) review the literature to identify potential flood models suited to application over large spatial and temporal scales, and iii) identify the most suitable flood model and apply to a large river valley. To successfully achieve these objectives, a series of principles were adopted to guide an iterative development of the model framework which was then stress-tested on the Gwydir River floodplain, New South Wales (NSW), Australia. These principles, in no particular order are: i) use NARCliM 1.5 outputs to force models suitable

for flood inundation estimation; ii) maximise benefit from inundation estimates by simulating the entire NARCliM 1.5 set of projections; iii) use open datasets, methods, models and mostly automatic approaches; iv) design the framework for implementation on high-performance computing resources; and v) the historical period, constrained by measurements, determines parameter values applied to the forecast period. The modelling framework that achieves our aim (Figure 1) and is consistent with these principles constrains both hydrologic and hydraulic models, takes boundary conditions from climate

model outputs, simulates them entirely by breaking them into four year windows with two month overlap for warming up the hydraulic model, develops initial conditions based on low flow conditions, simulated in parallel on high performance computing resources and has data management to limit the file size associated with saving inundation depth and speed by storing the daily maximum inundated depth and associated flow speed. The various segments are then combined (removing the two months overlap) and stored in compressed netCDF files (https://doi.org/10.48610/d7b1654). The hydrologic and

hydraulic methods used in this framework are discussed in Sect. 2.1 and 2.2.





## 2.1 Evaluation of climate model outputs and hydrological model theory

The NARCliM 1.5 climate model ensemble includes three global climate models (CCCma-CanESM2, CSIRO-BOM-ACCESS 1-0 and 1-3) with two regional climate models (UNSW-WRF360 J and K) resulting in a set of six model combinations (Nishant et al., 2021). Projections for two epochs (historical 1950 to 2005 and projections 2006 to 2099) using

two global emission pathway scenarios (RCP 4.5 and 8.5) are available, and include hourly variables of precipitation and total run off, and bias-corrected daily precipitation. NARCliM 1.5 was applied by matching, as much as possible, measured and modelled climate statistics. For catchment runoff, this was done at Gravesend on the Gwydir River, where the measured distribution of annual maximum discharge was used to calibrate the hydrologic model. Gravesend (figure 2) is the last gauging station before the conversion between water level and river flow rate becomes significantly uncertain (tailwater and

inundation feedbacks leading to significant hysteresis). Each of the historical river flow projections were calibrated using the measured distribution of annual maximum discharge at Gravesend. The hydrologic model used the excess precipitation (excess rainfall) obtained from NARCliM 1.5 ('total run off' code named mrro) in the following manner. The bias corrected daily rainfall was used to bias correct daily total run off (or excess daily rainfall), and this was interpolated onto an hourly timeframe using the NARCliM 1.5 hourly precipitation for shape. That is

$$\text{daily runoff corrected} = \text{daily runoff} \times \frac{\text{bias corrected daily precipitation}}{\text{daily precipitation}} \tag{1}$$

and

$$\text{hourly runoff on day } t = \text{daily runoff corrected on day } t \times \frac{\text{hourly rainfall on day } t}{\sum_{\text{day } t} \text{hourly rainfall on day } t} \tag{2}$$

where the last term in equation (2) ranges from zero to unity.

The excess precipitation was routed through catchment models following the method proposed by Mein et al. (1974), which is referred to as a ROR-style model with two free parameters, $m$ and $k$, that are nominally for flow shape and storage, but experience with this model indicates their theoretical basis is weak and they are used as free calibration parameters. The external catchments draining to the hydraulic model (figure 2) come from Gwydir River, Boggy Creek, Waterloo Creek, Curley Creek, Tycannah Creek, Mosquito Creek, North Creek and un-named watershed. Each catchment was broken into

between five and 13 sub-catchments, yielding an outflow suitable for use in the hydraulic model. The hydraulic model covers a significant area (9,621 km$^2$) and consequently, runoff onto the hydraulic model is included by associating sub-catchments with model grid locations. The climate projections have more than one grid cell within sub-catchments in many places, with these contributions reduced by the area of each cell from the climate projection overlaps each sub-catchment, with these contributions allocated in proportion to the grid cell overlap on the sub-catchment.

Comparisons with measurements of river flows at Gravesend, on a distribution basis, indicated that using NARCliM 1.5 to provide excess rainfall and a ROR-style runoff routing model with no losses (initial or continuing) leads to overestimates of frequent events and underestimates of infrequent events. This indicates that there is not enough loss of water volume during





lighter rainfall events compared to heavier rainfall events with in NARCliM 1.5. There are many on-farm water storages not included in the NARCliM 1.5 or catchment hydrologic models used to this point. To include them, we extended the

hydrology models by adjusting the excess precipitation before it is used for runoff routing. The excess precipitation was routed through a storage of maximum depth $h_{max}$, a surface area of $fA$ (where $A$ is the catchment area) while water within that storage was evaporated using monthly mean of measured evaporation rates and a usage rate to model farm use. The storages were initially started at half full. If the storage does not overflow during a time step, there will be no excess rainfall. If the storage does overflow, then there will be excess rainfall, $P_r$, to yield runoff. Mathematically, if $h$ is the depth of water in the

storage, then it will change by

$$\Delta hf = (P - (e + u)f) \times \Delta t \tag{3}$$

where $P$ is the NARCliM 1.5 excess precipitation, $e$ and $u$ are evaporation and usage rates and $\Delta t$ is the time increment. This adjustment was applied as follows:

$$h(t)f + \Delta hf > h_{max}f \qquad \begin{cases} P_r = \frac{fh + \Delta hf - h_{max}f}{\Delta t} \\ h(t + \Delta t)f = h_{max}f \end{cases}$$

$$0 \le h(t)f + \Delta hf \le h_{max}f \qquad \begin{cases} P_r = 0 \\ h(t + \Delta t)f = h(t)f + \Delta hf \end{cases} \tag{4}$$

$$h(t)f + \Delta hf < 0 \qquad \begin{cases} P_r = 0 \\ h(t + \Delta t)f = 0 \end{cases}$$

and if $f = 0$, then the model simplifies to $P_r = P$.

## 2.2 Selection of hydraulic theory and code

Climate change evaluation at regional scale or larger for flooding hazard and other applications requires fast and accurate

enough flood modelling. This review seeks to identify hydrodynamic models with proven track records to achieve this evaluation in a timely manner with limited human resources (automated processes). This assessment is separated into physics-based models and rules-based models.

Physics-based models typically follow Newton II and in particular, the shallow water equation or dynamic wave equation, applied in either one or two horizontal dimensions (e.g., 1D or 2DH), to solve for temporal and spatial variation in flow

depth and speed. There are several well-known approximations of the dynamic wave equation, with kinematic, diffusive, and partial inertial wave (or long wave) approximations possibly the best known. All physically based methods except dynamic wave exclude convective acceleration and hence, momentum changes required to change flow direction. Consequently, forces from water surface gradients required to get flow through geometry changes (road embankments across a floodplain) is reduced when compared to including convective acceleration. These terms have been found essential in ocean models

where mean water level gradients are exceedingly small and flow mass exceedingly large (mean ocean depth is ca 4 km). There are too many examples of successful dynamic wave application in two dimensions or combination of one and two dimensions to list them all, however the following subset (e.g., Montanari et al., 2009; Ahmadisharaf et al., 2018) highlight



methods aimed at accelerating applications for flood management including Graphics Processing Unit implementations through to careful use of 1D/2DH modelling (resulting in global scale continuous simulations). This approach remains the
benchmark theory for flood modelling.

Examples of successfully applied partial inertial wave models are numerous (e.g., Rajib et al., 2020; Sampson et al., 2015; Bates et al., 2010) and this approach has a proven track record of: statistical evaluations, hazard mapping or Monte Carlo Risk evaluations including damage estimations with velocity and depth contributions (e.g., Hoch et al., 2017; Neal et al., 2013), large spatial and temporal scale assessments where channels were sub-grid features (O'loughlin et al., 2020;
Schumann et al., 2013), multi-channel assessments (Altenau et al., 2017), temporal scales from minutes to years (e.g., O'loughlin et al., 2020; Neal et al., 2011) and on to geological scales (Coulthard et al., 2013), coastal storm surge inundation (Lewis et al., 2011), coastal tidal dynamics (Skinner et al., 2015), flooding in urban, rural, remote and limited data applications (e.g., Amarnath et al., 2015; Bates et al., 2010; Fewtrell et al., 2011; O'loughlin et al., 2020). While there are notes of caution with this approach at large scale (Schumann et al., 2012) and other authors advocating for the diffusive
wave (Dottori and Todini, 2013) over partial inertial wave, it has the best track record after the dynamic wave equation while being exceptionally quick. The partial inertial wave equation has a theoretical limit in that at either high velocity (Froude number exceeding 1) or low frictional force, the momentum equation becomes unstable. This well-known issue has been noted in the recent literature with respect to LISFLOOD.

The diffusive wave equation has a long track record dating back to when hydraulic modelling using numerical methods in
two dimensions started in the 1970's. However, in more recent times where it has been revisited for its light computing load (e.g., Mason et al., 2009; Apel et al., 2009; De Roo et al., 2000), it has been the reason for shifting to partial inertial wave equation (Neal et al., 2012), with only one reference found arguing diffusion over partial inertial wave (Dottori and Todini, 2013) for accelerated flood assessments. Further, there is evidence that diffusive wave does not handle urban environments (Costabile et al., 2017) but away from these areas and with enhancements, it is accurate enough (Jamieson et al., 2012). The
diffusive wave model links forces to motion exclusively through the friction model whereas the partial inertia wave model has a combination of friction and temporal acceleration. This fixed link through the adopted friction model means uncertainties in the friction model and spatial and temporal parameter variations are more significant in diffusive wave estimations. As the earlier engineers/scientists knew, applying diffusive wave theory to subcritical flow on a two-dimensional horizontal grid is often numerically unstable leading to the checkerboard predictions. While some recent authors
were seeking to address this numerical stability issue using careful spatial and temporal selections and flux gradient limiters, ultimately the decision to include the additional temporal acceleration (inertial) term resolved their numerical issues almost entirely. From the balance of evidence and theorical arguments, it is proposed that diffusive wave is an unacceptable approach when trading-off between accuracy and speed.

The kinematic wave equation has a long track record in modelling supercritical flows (Miller, 1984) with more limited
application to subcritical flow modelling of prismatic channels (Zheng et al., 2020). When the continuity equation is combined with the kinematic wave equation, predictions exclude flow attenuation and actually increase flow rates and water





surface slopes (Miller, 1984, page 18). In the case of prismatic channels, the water depth and flow rate are fixed or $Q = Q(h)$, where $Q$ is flow rate (Henderson, 1966, page 367) and yet numerical models of prismatic channels rarely achieve this and degrade to $Q$ increasing with both time ($t$) and position. Miller (1984, page 20) further indicates that for a successful
kinematic wave application, ad hoc modifications in how this equation is solved is required and then only on the rising limb. Consequently, large errors are expected when using kinematic wave equation in non-prismatic channel systems. The balance of evidence and theorical arguments indicates that kinematic wave equation is an unacceptable approach when trading-off between accuracy and speed.

The impact cell method is based on rules around how floodplains fill with water during flooding either over defences or by
defence failure. They use a dynamic wave equation one dimensional model to drive the floodplain filling and while they appear to be temporal, they are quasi-steady (Lhomme et al., 2009; Gouldby et al., 2008; Hall et al., 2003). The major drawback is model development in that it involves a combination of physical and probabilistic input, which have no apparent automatic techniques for their estimation. There is a lack of track record around estimating velocities from the water level gradients this style of model predicts.

The cellular automata method is based on a set of rules that mimic continuity and kinematic limits, which from limited testing (e.g., Jamali et al., 2019; Guidolin et al., 2016; Nicholas et al., 2006) is able to simulate urban areas, multi-channel systems, and hydraulic structures within a gridded domain. Various versions do include storage attenuation. There is, however, no track record around estimating velocities from the water level gradients this style of model predicts.

There are other rules-based methods including rating curve GIS models (e.g., Zheng et al., 2018; Apel et al., 2008) through
to dynamic and rule based combined models (Bernini and Franchini, 2013; Jamieson et al., 2012). These have not been considered as they exclude flow routing.

The trade-off between accuracy and computational effort and seeking flood hazard information thereby requiring reasonable flow speed estimates, leads to the selection of partial inertial wave equation (LISFLOOD) and the cellular automata (WCAD2D). These two hydraulic models were tested for speed and LISFLOOD was found to be 2 to 2.5 times faster when
tested on large floodplains such as the Gwydir River. This led to the selection of LISFLOOD.

**2.3 Implementation of LISFLOOD hydraulic model**

Surface roughness (using Manning's $n$) for the LISFLOOD model developed here was obtained from existing calibrated hydraulic models for the Gwydir River. There are three models forming the NSW Department of Planning and Environment Gwydir River hydraulic model with 1D links and 2D grids with resolutions from 20 m to 50 m using MIKE FLOOD
(Anonymous, 2015) (NSW Department of Primary Industries, Water 2015). After balancing resolution with file size and run times, a 100 m resolution was selected. These three models were combined to develop the 100 m DEM with extents to enclose Binniguy to Moree, Moree to Barwon and Thalaba Creek MIKE FLOOD hydraulic models (colour shaded area in figure 2). The origin was set so that the 100 m DEM collocated with every second grid point of the Moree to Barwon model. Crest features (usually roads, but any feature that could either act as a weir or dam that changes flow distributions) were



extracted out of Binniguy to Moree, Moree to Barwon and Thalaba Creek DEMs, and put onto the 100 m DEM. This was
achieved in a two-step process, first a smooth version of each existing DEM was subtracted from the new 100 m DEM and
differences below 0.2 m removed. The resulting features showed crests as well as other differences related to waterways. The
crest features alignments were then determined, and the crests extracted. Waterways removed from Binniguy to Moree were
put back in using survey DEM, missing areas were filled in using Shuttle Radar Topography Mission data and finally,
streams were hydraulically connected (figure 2).

The three hydraulic models forming the Gwydir River Hydraulic model by NSW government was used to constrain the
LISFLOOD model, using their 2012 calibration runs, performed in MIKE FLOOD. There are complications in that those
NSW government models included 1D elements, had finer resolution (20 m and 50 m) and were separated into three
domains, one run in steady state (southern region) and the other two using dynamic simulations with varying simulation
periods; compared with the one encompassing LISFLOOD model, which had a coarser resolution (100 m) and no 1D
elements. To rationalise these comparisons, locations where the NSW government models had reported inundation were
used to constrain the LISFLOOD model. The first calibration series ran 100 incremental model topographies from largest
main channels possible from survey to no channels, and inflows taken directly from the NSW government models. The
channel geometry was selected to obtain the best match to these calibrated models.

**2.4 Climate Projection to flood simulations**

NARCliM 1.5 includes six historical projections and 12 future projections providing 18 periods for simulating, covering a
total physical time of 1,470 years. Such simulations require high performance resources and careful selection of outputs and
model resolution to ensure simulations are obtained within a reasonable timeframe. Within storage resources available,
output from LISFLOOD was hourly and then postprocessed to daily information of maximum inundation depth and the flow
speed at that maximum depth. This, with several storage techniques to minimise file sizes (netCDF with compression and
finite data resolution), reduced required storage from ca 10 TB to 100 GB. Applications involving steeper catchments and
floodplains may warrant storage of hourly rather than daily outputs. To further enhance model throughput, simulations were
broken into four-year segments, with an additional two-month warmup period using initial conditions taken from a low flow
simulation developed from measurements and average evaporation. The two-month warmup period was confirmed to not
impact predictions by comparing predictions from the end of a segment with the predictions (after warmup) at the start of the
following segment. The model grid was selected after initial testing of four resolutions of 50 m, 75 m, 100 m and 150 m.
These tests indicated that simulation times, from finest to coarsest grids was 55, 16, 7 and 2.5 days per decade respectively,
while mean biases from the 50 m resolution were 1 cm, 5 cm and 12 cm for the 75 m, 100 m and 150 m resolutions grids.
The 100 m grid was a reasonable balance between output size, simulation speed and model performance for resources
available. That is, a reasonable balance between loss of accuracy cf 50 m and 75 m resolution when compared with eight-
and two-fold decrease in computational resources. The LISFLOOD version implemented was the latest available at the time


(February 2021), compiled with the 2018 version of Intel C++ and ran on CentOS version 7. These simulations took several weeks using high performance computing resources where between 160 to 480 threads were available.

## 2.5 Flood hazard classes

The flood hazard classification shown in figure 3 (Smith et al. (2014b) is recommended for use in emergency planning and management within Australia (Ball et al., 2019) and has been applied here. The classification has six classes, starting with the safe classification H1 (generally safe for vehicles, people and buildings) through to H6 (unsafe for vehicles and people and all building types considered vulnerable to failure). In applying these flood hazard classifications, one additional hazard classification was added to capture flood hazards exceeding the maximum class (H6). Additionally, regions with no
inundated areas over the analysis period were assigned to the safe hazard class H1.

## 2.6 Bernoulli's trial to assess flood hazard class changes

The assessment of climate changes on flood hazard classification had to deal with a range of climate model projections spanning dry through to wet which have significantly different flood projections and associated flood hazards. Consequently, each flood hazard classification was treated separately, and assessments were done on an annual basis for a historical epoch
of 1980 to 1999, and projected epochs of 2020-2039 for near-term, 2050-2069 medium-term and 2080-2099 for long-term comparisons. The occurrences of each flood hazard classification are then the number of times it occurs divided by 20, the number of years within these epochs, which is a maximum likelihood estimate of the occurrence probability given 20 independent binomial (Bernoulli's) trials. Once occurrence probabilities are known for each epoch in each flood projection, they are averaged or ensembled across the flood projections from the six climate model combinations before estimating
changes between epochs.

## 3 Results

### 3.1 Calibration of hydrologic model

The hydrologic model calibration to annual maximum discharge at Gravesend (figure 4) was achieved using the same $m$ (nominally stream shape, which is expected) and different $k_c$ (channel storages) and the same small catchment storage
parameters ($f = 0.0005$, $h_{max} = 0.2$ m and $u = 80$ mm/day) across the six historical climate projections available in NARCliM 1.5. Uncertainty remains with the adopted calibration, which is minimised for inundation hazard assessment by focusing calibration on rarer events.




## 3.2 Calibration of hydraulic model

The hydraulic model, LISFLOOD, was calibrated by varying the main channel depths until it matched previous models,
MIKE FLOOD, that had been calibrated to historical floods. The hydraulic model with channel depth at 19% of the
maximum channel depth had mean flood level differences of less than 1 mm (figure 5, top left panel) while also being near
the lowest standard deviation of flood level difference. As LISFLOOD and MIKE FLOOD models had different resolutions
and consequently different ground surface elevations, comparing depths bring in two changes, one related to hydraulic
performance and another related to ground surface elevation interpolation differences (figure 5, bottom left panel).
Alternatively, comparing water surface levels (figure 5, top right panel) removes this ground surface elevation interpolation
aspect, however, for models with large vertical variation (e.g., Gwydir River has 100 m vertical change over its 167 km
length), this vertical variation overpowers water level differences when plotted. Nevertheless, comparing differences of both
depth and water surface level together with an overall water level difference map (figure 5, bottom right panel), provides a
visual assessment of model calibration.

## 3.3 Flood hazard classification and changes under RCP 4.5 and RCP 8.5


The occurrence probabilities under both RCP 4.5 and 8.5 (figure 6, table S1) for flood hazard classification H1 (generally
safe for people, vehicles and buildings) are predicted to increase while higher hazard classifications (generally dangerous for
people, vehicles and buildings) are predicted to reduce in the long-term (comparing 2080—2099 with 1980—1999) for the
NARCliM 1.5 ensemble. Within this ensemble, the H1 occurrence probability changes for RCP 4.5 vary from no change to
an increase of 0.3 and for the RCP 8.5 increases from 0.06 to 0.39 (figures S1-S6), indicating high likelihood of a reduction
in flood hazard at the valley scale. This longer-term assessment outcome does not apply for the near- or medium-term
(2020—2039 or 2050—2069, table S1). The change expected in the near-term are very slight (increase in H1 by 0.01 to
0.02) but the ensemble includes projections where the H1 occurrence probability is reduced by 0.09. These decreases in H1
come with increases in H2 through to H4 of between 0.03 to 0.13. The medium-term comparison period is a transition
between the other two with RCP 8.5 always increasing H1 and decreasing H2 through to H4 and RCP 4.5 having both
increases and decreases of H1 through to H4 within the ensemble.

## 4 Discussion

The increases in H1 occurrence coupled with decreases in H2—H4 (figure 6, table S1 and figures S1-S6) indicates that flood
hazard is decreasing in the long term under projected climate changes (all cases in the ensemble and both RCP 4.5 and 8.5)
in the region modelled (figure 2). The near-term changes are more uncertain as there are cases in the ensemble that both
increase and decrease flood hazard (table S1). Comparing near-, medium-, and long-term, RCP 8.5 shows more certain
decrease in flood hazard compared with RCP 4.5, however, in both scenarios, the most likely outcome is a decrease in flood
hazard with all members of the ensemble forecasting this.





The inference that flood hazard is decreasing in this region with projected climate change comes with several key limitations. Hydrology models were calibrated to best represent infrequent events across the historical period. Consequently, these models overestimate the catchment runoff from frequent events by different amounts for each member of the ensemble (figure 4). These differences come from the climate models themselves where the rainfall runoff is estimated using different approaches leading to different outcomes across the one historical period. That is, the distribution of runoff of each member

of the ensemble for the historical period, in the absence of epistemic uncertainty, should be similar. Whereas these distributions are different and consequently, add to the uncertainty of inundation depth and speed projections, both used to assess flood hazards. The hydraulic model, which was constrained reasonably given the differences between resolution and modelling approaches (figure 5), is less an issue compared with hydrologic uncertainty. However, there is still differences between estimates (figure 5) from various flood projections that may lead to different conclusions spatially. Finally, when

estimating changes in flood hazard, this would usually involve estimates of flood hazard under extreme conditions. However, the assessment provided used an alternative method for reasons discussed in the following paragraphs.

Conventional extreme value analysis for flood hazard assessments involves establishing a link between flow discharges and exceedance probabilities. This relationship then can be used to assign exceedance probabilities either to historical events or

synthetic events that represent historical events, which are simulated, and the spatially varied maximum flood hazard obtained. This approach would work for systems that are driven by one major inflow and have flooded area relatively small compared to the rainfall systems that excite flooding. However, the floodplain being assessed has a large catchment area compared to the spatial size of rainfall events and while it has one major inflow, there are several others, and those combined with the floodplain itself, makes breaking continuous simulations into a series of events where the probability is constant

across the floodplain inundated area, a subjective (or arbitrary) assessment.

Another issue in using conventional extreme value analysis for flood hazard assessments is the balance between projection period and ability to establish reasonable extreme value estimates. For example, one can do a simple numerical experiment in which the two distributions are constructed with a fixed increase in all extremes (simplest case), and then draw one

sample, the estimated extreme values, obtained from fitting to this sample, can be both an increase or decrease compared with that assumed and this is due to sampling error when the analysis period is smaller than the extreme value return period being estimated. To robustly estimate an extreme value, using a one-off sample, the analysis period usually needs to be many times its return period (rule-of-thumb, 10 or more). Without this, the sampling error overwhelms any changes and thus any changes that are within the confidence limits are statistically insignificant.


The final issue in using conventional extreme value analysis comes from the differences in inundation extents and frequency across the climate model predictions that span dry through to wet conditions. This led to significant areas which were





inundated in the wettest projections that remained dry in the driest projection. Consequently, the members within the ensemble would vary spatially, making uncertainties difficult to understand and communicate.


Applying extreme value theory to individual grid inundation flood hazard (i.e., linking exceedance probability directly to flood hazard, after applying either block maxima or peak over threshold approach on independent and identical distributed events to determine extreme events), as opposed to the conventional method of linking probabilities through event peak discharge, is that the number of extreme events changes from many events along deep watercourses to approaching zero near

the edge of maximum inundation. This variation of number of extreme events lead to reasonably consistent spatial predictions along deep watercourses to inconsistent spatial predictions across the floodplain were number of extreme events approaches zero at the edge of inundation. These spatially inconsistent predictions were obtained for a range of extreme value approaches and fitting methods. Furthermore, near the edge of maximum inundation, the extreme value models themselves broke down as the number of events approaches zero. The net result being very limited consistency in linking

exceedance probabilities to flood hazard across the floodplains, particularly near the edge of maximum inundation.

Our approach (sections 2.5 and 2.6), where we estimate changes in annual probability of occurrence of flood hazard classes overcomes issues with conventional and grid based extreme value analysis.

## 5 Summary

A modelling framework for estimating projected flood hazards from regional climate model projections has been presented including a different approach to assessing flood hazard changes. The modelling framework was applied to Gwydir River (Australia) using New South Wales and Australian Capital Territory Regional Climate Modelling version 1.5 projections with computationally efficient hydrologic and hydraulic models. This included six historical and 12 future regional climate projections. The simulations were continuous and totalled 1,470 years, requiring high-performance computing resources for

timely completion. The climate projections included spatially varied rainfall runoff, allowing the implementation of a hydrological modelling approach that only required flow routing as soil dynamics were included in the regional climate models. The hydrology model was constrained by measured distributions of runoff. The hydraulic modelling approach was selected after an extensive evaluation and testing phase of modelling types with proven track records of computational efficiency, leading to the selection of the partial inertial wave equation as implemented in LISFLOOD over the other family

of efficient approaches under the cellular automata umbrella. This hydraulic model was constrained by modifying the main channel geometry until it matched more detailed and calibrated hydraulic models using the dynamic wave equation. The simulations resulted in spatially varied daily maximum flow depth and flow speeds at those depths across the 18 regional climate projections, allowing flood hazard assessments.



Changes in flooding hazard were assessed by estimating changes in the annual probability of occurrence of a range of flood hazard classes, with the first class, H1, being a safe class and all other classes having various levels of flooding hazard. This approach was taken to overcome several barriers in using conventional flood hazard assessment techniques where flooding hazards are estimated at various extreme values. These barriers included variable number of hazard events across the floodplain, the ability to determine an extreme value where the underlying processes are changing through to regional

climate projections ranging from dry to wet leading to significant differences in inundation extents. Changes in annual probability of occurrence in the long-term are consistently, across the ensemble for both RCP 4.5 and 8.5, indicating a reduction of flooding hazard across Gwydir River region modelled. This was demonstrated as increased probability of occurrence of the safe class (H1) and decreased probability in all the unsafe classes. The outcomes are more mixed in the near-term, with the ensemble indicating minor decreases in flooding hazard albeit with ensemble members having both

increases and decreases. The medium-term projections are transitional between the near- and long-term, however, there remains ensemble members with increased flooding hazard.

*Code and data availability*. https://doi.org/10.48610/d7b1654

*Author contribution*. DC led methodology, software development, model validations and visualization and formal analysis with substantive contributions in each of these by MH, MH led conceptualization, resources (climate model forcing), project administration and project funding acquisition, DC and MH contributed to writing, both original draft preparation, reviewing

and editing.

*Competing interests*. The authors declare that they have no conflict of interest.

*Acknowledgements*. The high-performance computing was supported by Queensland Cyber Infrastructure Foundation and

The University of Queensland. This project was financially supported by the NSW Climate Change Fund. The support from Matthew Riley and Tim Pritchard in initiating the project is greatly appreciated.

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



Hydrology:-

NARCliM 1.5 historical outputs → Hydrology modelling → Flow estimates at gauging station(s) → Compare modelled & measured flow distributions → Establish hydrology model inputs

Hydraulics:-

Previous Hydraulic Model Predictions for a particular event → Hydraulic modelling → Constrain inundation predictions to match previous predictions → Establish inundation model parameters

Simulations:-

Apply hydrology for boundary conditions → Run low flow case to establish initial conditions → Break simulation period into 4-year segments → Run each segment in parallel using high-performance computing resources

Write to binary files and compress using tar ← Process during runtime to obtain daily maximum depth and flow speed at that depth ← Record hourly water depth and flow speed

Once all runs completed combine segments and convert to compressed NetCDF → Post processing activities (checking and usage)


**Figure 1: Proposed framework for converting climate model outputs to flood model outputs.**

Natural Hazards
and Earth System
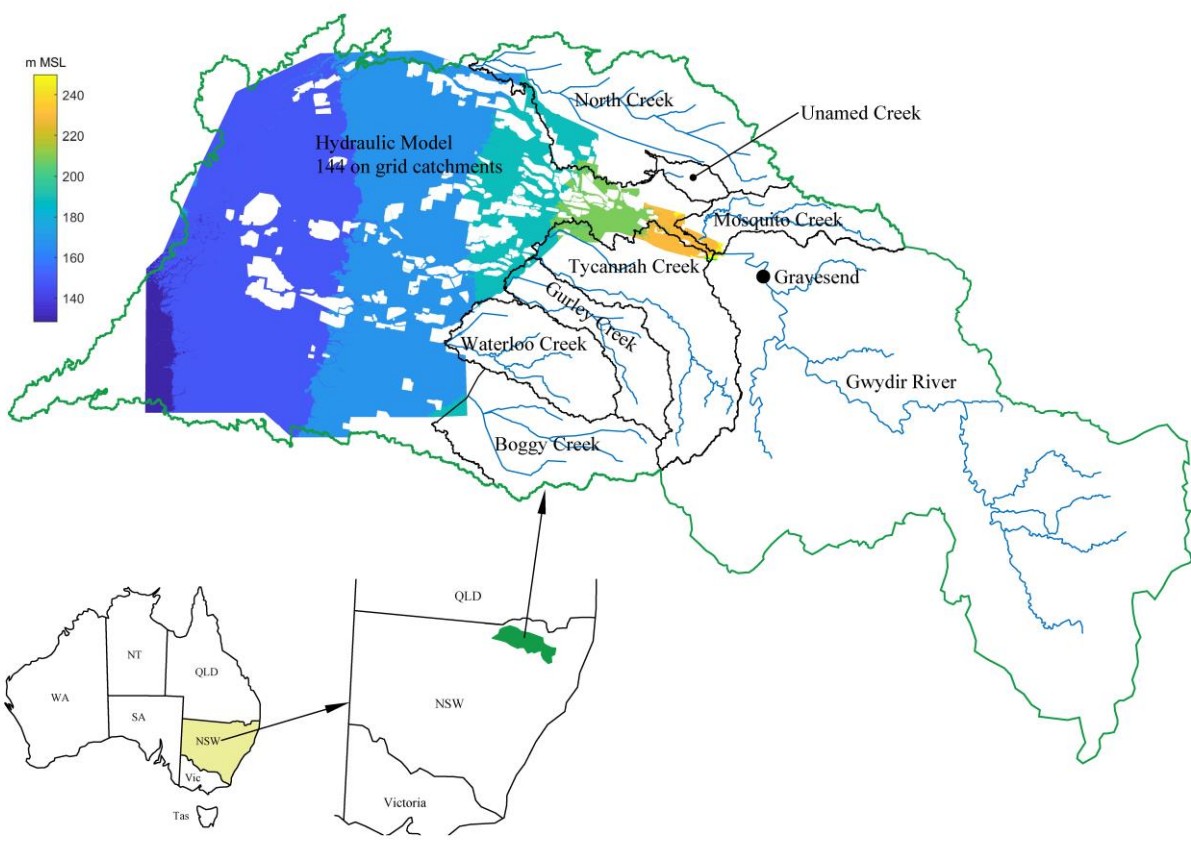

**Figure 2 Catchments and waterways flowing through the Gwydir Valley with the location within New South Wales (NSW), Australia shown in the bottom left inserts. Hydraulic model extents shown by colour shaded area representing ground elevation in metres above mean sea level (colour bar) with the main source of inflows from the Gwydir River, which has a gauging station at Gravesend (●). The 133 watershed boundaries within the hydraulic model and sub-catchments within each waterway not shown for clarity.**


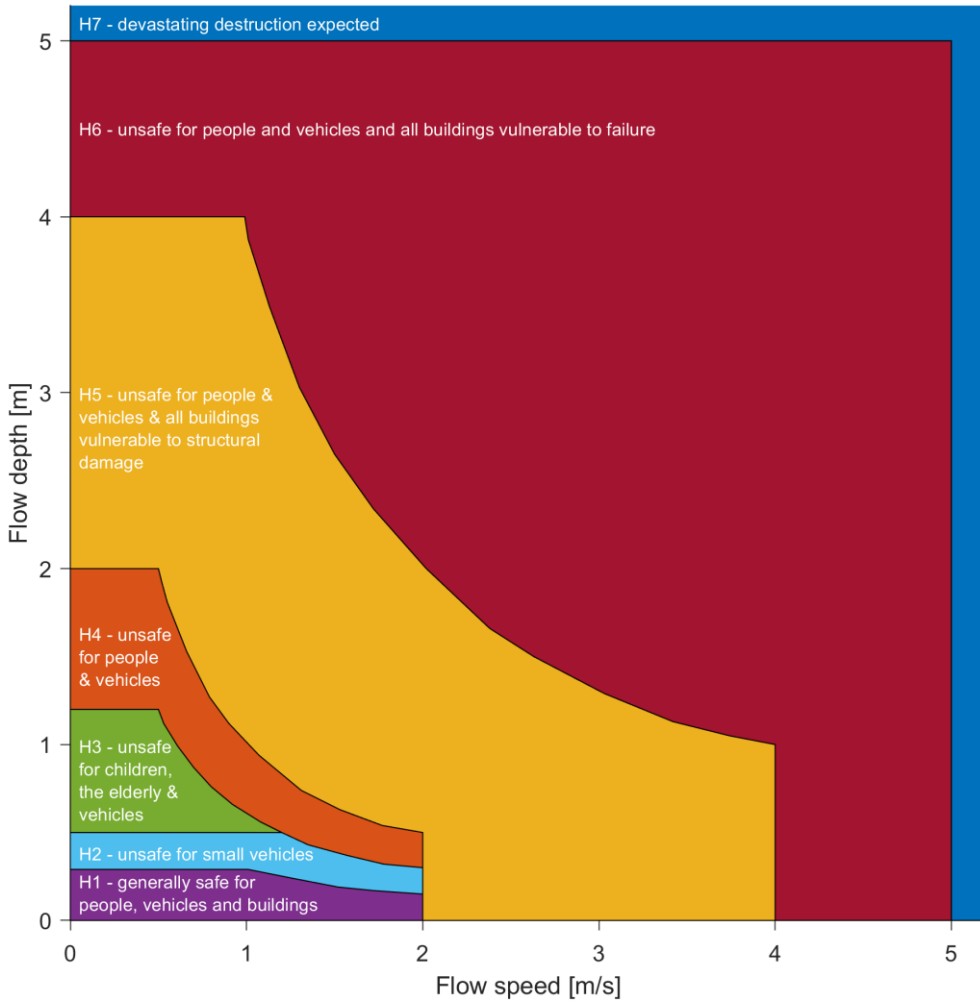

**Figure 3 A flood hazard classification scheme from H1 (safe) to H6 (dangerous) recommended for use in Australia. Flood hazard class H7 is additional to the recommended classifications.**

**Figure 4 Comparisons between modelled discharge at Gravesend (figure 2) with measurements for all global (rows) and regional (columns) climate model combinations. Model discharges includes additional evaporation via local storages ($f$ = 0.0005, $h_{max}$ = 0.2 m and $u$ = 80 mm/day). Hydrology model parameter $m = 0.5$ for all cases and $k_c$ is indicated in each panel. The black line indicates perfect agreement, the solid-coloured line and corresponding shaded region are mean and 95% confidence of measured distribution when resampled to compare with modelled discharges. Difference colours indicate different $k_c$.**


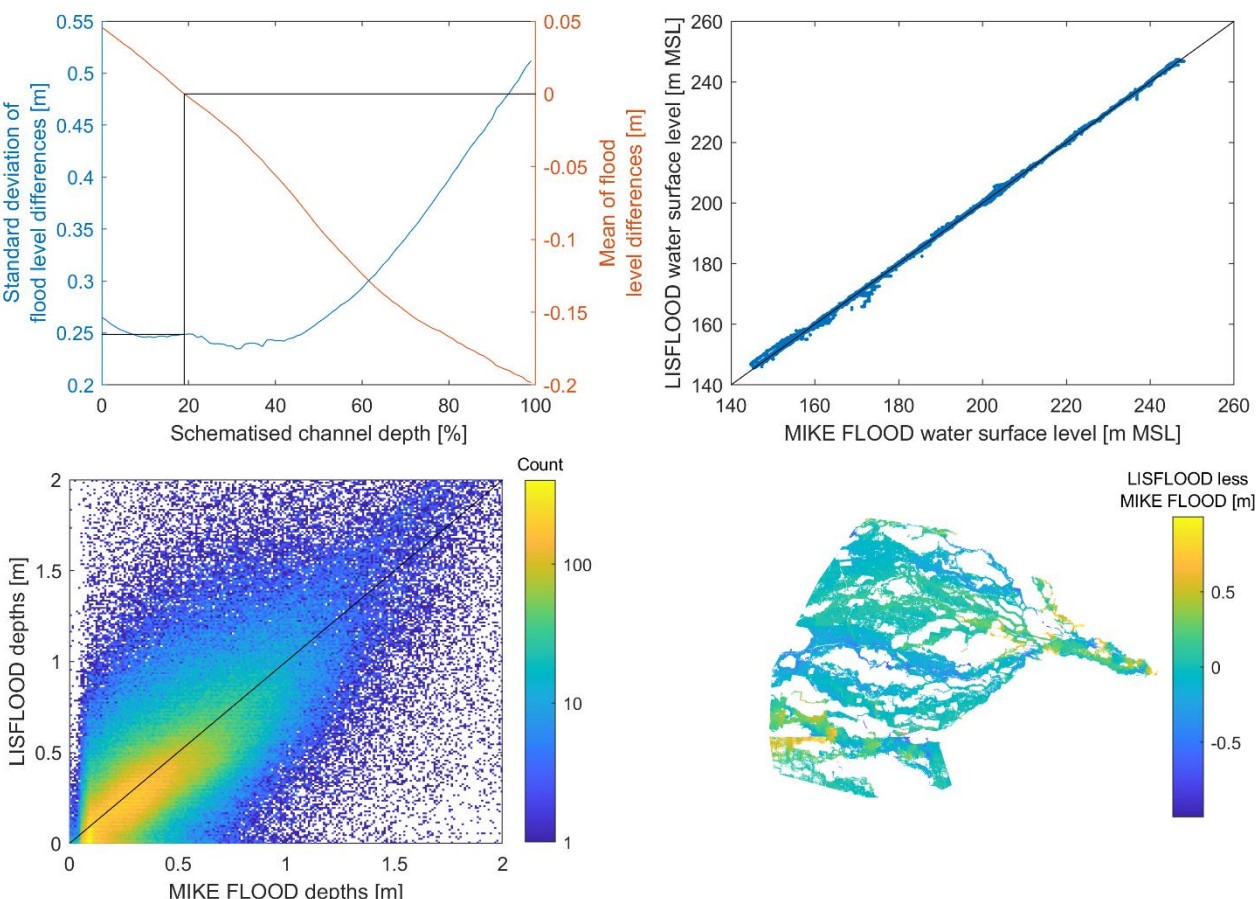

**Figure 5 Gwydir River hydraulic model (LISFLOOD) calibration to existing MIKE FLOOD hydraulic models by NSW Environment & Heritage. Top left panel, selection of schematised channel depth (zero means no channels and 100% means largest main channels possible from survey) with the black lines showing selected channel depth. Top right panel, a comparison between flood levels across the entire model (blue dots) with perfect fit (black line). Bottom left panel, a comparison between flood depth across the entire model. Bottom right, difference map between models.**



**Figure 6 Gwydir Valley Flood hazard historical (1980—1999) classification occurrences and their changes under RCP 4.5 and RCP 8.5 (2080—2099) for the NARCliM 1.5 ensemble. The mean of occurrence probability changes, $\delta$, shown in each panel. For brevity, flood hazard historical classifications H5 to H7 are not shown as they are limited to within river and creek channels.**