# Peer review of "Assessing flood hazard changes using climate model forcing"

_Natural Hazards and Earth System Sciences, 2022_

## Author Response (AR1)

In the following, all line numbers provided by the reviewer are for the original submission and those provided by us are for the corrected manuscript.

While undertaking these corrections, a number of minor errors (e.g., were when where should have been used) has also been correction (see tracked changes).

Review 1

The fact that the hydrology model was a flow routing model (no infiltration) could have perhaps been introduced earlier in the text and Figure 1.

Corrected. The rainfall-runoff routing model is introduced in the first section of section 2, which has been modified to explicitly state the infiltration is taken into account in the climate models by adding at line 94, "models that use rainfall less that used by infiltration". Additionally, within the introduction (line 83) where the hydrology model is briefly introduced, we change "ROR-style hydrology model" to "hydrological flow-routing model" and edit "hydraulic model with climate projections are described, driven by NARCliM1.5 climate projections as an example" to "hydraulic model with climate projections for rainfall-runoff (or excess rainfall). NARCliM1.5 climate projections are used as an example".

What is the "bias correction" in the regional models correcting and how big do these biases get?

Correct. Bias correction scales the climate projections for the past to improve the fit with historic observations thereby accounting for systematic errors in the projections. The creators of the NARCliM projections had undertaken bias correction for daily precipitation but not for daily or hourly runoff. Since the bias correction process is detailed and not central to this article, we provided a citation to an article that addresses this question. The change to address this comment is adding the following text after the first time "bias-corrected daily precipitation" occurs (line 115) of "(corrected to observed precipitation distribution; see e.g. Evans et al. 2021). While NARCliM 1.0 selected CMIP3 GCMs, NARCliM 1.5 selected CMIP5 GCMs from the unsampled space within NARCliM 1.0, all with similar temperature increases but spanning the range of precipitation changes from no change to moderate decrease to large decrease (Nishant et al., 2021)."

This correction adds the follow two references.

Evans, J. P., Di Virgilio, G., Hirsch, A. L., Hoffmann, P., Remedio, A. R., Ji, F., Rockel, B., and Coppola, E.: The CORDEX-Australasia ensemble: evaluation and future projections, Climate Dynamics, 57, 1385-1401, 10.1007/s00382-020-05459-0, 2021.

Nishant, N., Evans, J. P., Di Virgilio, G., Downes, S. M., Ji, F., Cheung, K. K. W., Tam, E., Miller, J., Beyer, K., and Riley, M. L.: Introducing NARCliM1.5: Evaluating the Performance of Regional Climate Projections for Southeast Australia for 1950–2100, Earth's Future, 9, e2020EF001833, https://doi.org/10.1029/2020EF001833, 2021.

I see that LISFLOOD is used for very large domains. Perhaps a sentence or two explaining why the hydrologic model is still warranted would fit.

We added the following explanation at the start of section "2.3 Implementation of LISFLOOD hydraulic model" near line 238.

"The LISFLOOD model was limited to the region covering the Gywdir River Floodplain of 8,100 km$^2$. LISFLOOD could have been applied across the entire catchment, removing the need for including a hydrology model. While this may be useful in particular situations, for the river valley being

investigated, that would require a LISFLOOD model grid covering 2.8 times more area, leading to significant additional computational resources. Consequently, the ROR-style hydrology model provides a trade-off between computational resources and framework complexity."

Figure 2. The colours of the bathymetry and the associated colour bar have me believing that the catchment is stepped.

Changed colour scheme to remove stepped appearance. New figure is now

[Figure]

Figure 2. What are the white patches? Could they be important?

The white patches are areas that have been leveed off for farming purposes. This has been included in the figure caption by adding "The white areas within the hydraulic model grid are grid points surrounded by levees and are unavailable to convey water." (line 603)

Line 278. The words "flow", "flow rate" and "discharge" are all used in this paper. Is discharge required?

We have changed to discharge throughout the manuscript. These changes are shown in the tracked changes document. We also took this requirement to be consistent to fix the other inconsistent language with projections and predictions was used interchangeably, has now been correction to "projection(s)" through out.

Line 231 and elsewhere. The word "constrain" is new to me in model development lingo.

The word "constrain" was used in the normal English sense, to mean this model was calibrated to existing calibrated hydraulic models. We addressed this comment by adding to the first occurrence of constrain (line 257), the following "(to previously calibrated hydraulic models)"

Figure 5 & 6. Wherever differences are plotted, I like the colour scheme to centre around white, with +ve value an increasing shade of red and -ve values a decreasing shade of blue, otherwise it is rather ambiguous.

The colour scheme was changed as suggested in both the manuscript and supplement, e.g.,

[Figure]

How many 1D structures were there in the original MIKE models and how big were they? I know we are doing comparisons here but we are also going to the effort of using a hydraulic model

There were no 1D structures in the original MIKE models. This has now been explicitly stated by adding "(channel links without hydraulic structures)" when introducing these MIKE models on line 245.

Reviewer 2

General Comments

•       It could be useful to better explain in the introduction the novelty of the paper since in literature there already are some articles that assess future flood hazard under climate changes scenario by using hydrologic and hydraulic models. In the present form the original contribution could be not so evident because it is not fully clear how the proposed methodology differ or increase its effectiveness from other studies on this topic.

•       I suggest in the introduction to add more recent bibliography on this topic and information about what was already proposed in other countries, i.e.:  1) Ryu, J.-H.; Kim, J.-E.; Lee, J.-Y.; Kwon, H.-H.; Kim, T.-W. Estimating Optimal Design Frequency and Future Hydrological Risk  in Local River Basins According to RCP Scenarios. Water, 2022, 14, 945, https://doi.org/10.3390/w14060945. 2) Shrestha, S.; W. Lohpaisankrit W. Flood hazard assessment under climate change scenarios in the Yang River Basin, Thailand. International Journal of Sustainable Built Environment, 2017, 6, 285–298, https://doi.org/10.1016/j.ijsbe.2016.09.006. 3) Janizadeh, S.; Pal, S.C.; Saha, A.; Chowdhuri, I.; Ahmadi, K.; Mirzaei, S.; Mosavi, A.H.; Tiefenbacher, J.P. Mapping the spatial and temporal variability

of flood hazard affected by climate and land-use changes in the future. Journal of Environmental Management, 2021, 298, 113551, https://doi.org/10.1016/j.jenvman.2021.113551.

The reviewer is correct in these two comments that we are not the first to use both hydrologic and hydraulics models when assessing flood risk changes from a changing climate. However, we do appear to be the first to use the runoff from climate projections, simulating them over the entire projection period to produce flood projections. Other work in this area uses climate projections to determine key events or design events and simulation of those are undertaken. That is, we first determine flood projections and then use these to assess risk changes. We propose adding a new paragraph at line 68 of

"Recent work investigating projected changes in flood risk under plausible climate futures includes Shrestha and Lohpaisankrit (2017) who forced a rainfall runoff model to estimate changes in discharges in the streamwise direction, allowing evaluation of changes in future risk. Moreover, Janizadeh et al (2021) trained a machine learning model to convert basin geometry and rainfall into risk, which was used with climate projections to evaluate future risk changes. Finally, Ryu et al (2022) analysed adjusted rainfall projections using flood frequency methods to assess risk changes at the basin level. The method here seeks to extend these by using a physics-based model to convert runoff into spatially explicit water surface levels and speeds across the entire floodplain and throughout the entire climate projection period. This objective overcomes issues around data poor regions (i.e., where machine learning methods are not possible), provides flood projections at consistent spatial and temporal resolutions across the full extents of the model (both streamwise and cross-stream), and permits application to river systems with complex hydraulics and discharge patterns (e.g. multiple and parallel channels) which rainfall-runoff models are unable to reasonably simulate."

And editing paragraph starting at line 80 as follows with italics showing changes

"The purpose of this paper is to describe the successful application of a modelling framework developed to convert climate model projections to hydrodynamic outputs, which were then used to assess future changes to present-day regional flood hazard. We demonstrate the utility of the approach by applying it to the Gwydir River, a large valley-floodplain system located in the northern Murray-Darling Basin, Australia. After reviewing candidate numerical models, new methods for driving *hydrological flow-routing* model and the LISFLOOD-FP hydraulic model with climate projections for rainfall-runoff (or excess rainfall). NARCliM1.5 climate projections *are used* as an example. *Rather than using the climate projections to determine key or design events for simulation, we simulated river floodplain hydraulics for the full climate projection time series.* Projected future regional flood inundation extents and the spatial distribution of flood hazard are presented for two global emission pathways (RCP4.5 and RCP8.5). Challenges associated with spatial and temporal sparsity in floodplain inundation and applying conventional extreme value distributions to evaluate future flood exceedance probabilities are discussed. These confound efforts to answer the question – will present-day flood hazard change under future climate projections – and we provide a new approach to answering that question."

Specific Comments

•       Lines 214-215: LISFLOOD was preferred to WCAD2D because it was found that the first model was faster than WCAD2D. Did you compare these model only for speed or also in terms of flood modelling results? In the latter case, did the test performed show significant differences?

The tests between LISFLOOD and WCAD2D involved similar water level estimates for both steady and unsteady tests, consistent with previous evaluations within the literature. As the article points out the LISFLOOD is considerably quicker, we edited lines 232 to 236 as follows with italics showing changes

"The trade-off between accuracy and computational effort and seeking flood hazard information thereby requiring reasonable flow speed estimates, leads to the selection of partial inertial wave equation (LISFLOOD) and the cellular automata (WCAD2D). These two hydraulic models *were compared in both steady and unsteady tests and evaluated* for speed. *While estimates of flood levels from the two models were similar*, LISFLOOD was found to be 2 to 2.5 times faster when tested on large floodplains such as the Gwydir River. This led to the selection of LISFLOOD."

•      Lines 241-242: please explain how you derive a total physical time of 1470 years starting by the 18 projections included by NARCliM 1.5.

The climate model ensemble includes six global-regional climate models that delivered 6 historical projections from the start of 1951 to the end of 2005, which is 55 years each and a total of 330 years. The climate model ensemble also delivered six future projections for each of two emission pathways. These 12 future projections from the start of 2006 to the end of 2100, which is 95 years each, total 1140 years. Consequently, the total from historical and future projections is 330 plus 1140 or 1470 years.

•      Paragraph 2.6 (Lines 267-275): I don't understand the criterium for selecting the epochs for flood hazard classification. How did you select as historical epoch the period 1980/1999, and as projected epochs the periods 2020/2039, 2050/2069 and 2080/2099 in the entire range 1950-2100?

Corrected. The periods selected follow those typically used for near, mid and far future time horizons in Australian government planning. At line 297, a new sentence will be added "These future epochs correspond to those typically used for near, mid and far future horizons in government planning."

Technical corrections

•      Line 13 pag. 1: historical period (1950-2006) should be the same of that one reported in line 99 (1950-2005).

The abstract and body of the manuscript have been corrected to match the projections to "(**1951-2005**) and a future period (2006-**2100**)".

•      Line 489 pag. 15: in the reference you miss probably the comma before 2009.

Corrected.